Optimised biomolecular extraction for metagenomic analysis of microbial biofilms from high-mountain streams

Busi Susheel Bhanu susheel.busi@uni.lu 1
Pramateftaki Paraskevi 2
Brandani Jade 2
Fodelianakis Stilianos 2
Peter Hannes 2
Halder Rashi 1
Wilmes Paul 1
Battin Tom J. 2
1 Systems Ecology Research Group, Luxembourg Centre for Systems Biomedicine, University of Luxembourg , Esch-sur-Alzette , Luxembourg
2 Stream Biofilm and Ecosystems Research group, École Polytechnique Federale de Lausanne , Lausanne , Switzerland
Nelson Craig
Electronic publication date: 2020 Oct 27
Publication date: 2020
Volume: 8
Electronic Location ID: e9973
Received 2020 May 4; Accepted 2020 Aug 26
Copyright: ©2020 Busi et al.
Copyright year: 2020
Copyright holder: Busi et al.
License: This is an open access article distributed under the terms of the Creative Commons Attribution License, which permits unrestricted use, distribution, reproduction and adaptation in any medium and for any purpose provided that it is properly attributed. For attribution, the original author(s), title, publication source (PeerJ) and either DOI or URL of the article must be cited.
License URL: https://creativecommons.org/licenses/by/4.0/

Keywords: Biofilms, Streams, Alpine streams, Glacier fed streams, Glaciers, Metagenomics, Biomolecular extraction

Funding: Swiss National Science Foundation CRSII5_180241 This work was supported by NOMIS Foundation and the Swiss National Science Foundation (CRSII5_180241). The funders had no role in study design, data collection and analysis, decision to publish, or preparation of the manuscript.

==============================
Glacier-fed streams (GFS) are harsh ecosystems dominated by microbial life organized in benthic biofilms, yet the biodiversity and ecosystem functions provided by these communities remain under-appreciated. To better understand the microbial processes and communities contributing to GFS ecosystems, it is necessary to leverage high throughput sequencing. Low biomass and high inorganic particle load in GFS sediment samples may affect nucleic acid extraction efficiency using extraction methods tailored to other extreme environments such as deep-sea sediments. Here, we benchmarked the utility and efficacy of four extraction protocols, including an up-scaled phenol-chloroform protocol. We found that established protocols for comparable sample types consistently failed to yield sufficient high-quality DNA, delineating the extreme character of GFS. The methods differed in the success of downstream applications such as library preparation and sequencing. An adapted phenol-chloroform-based extraction method resulted in higher yields and better recovered the expected taxonomic profile and abundance of reconstructed genomes when compared to commercially-available methods. Affordable and straight-forward, this method consistently recapitulated the abundance and genomes of a mock community, including eukaryotes. Moreover, by increasing the amount of input sediment, the protocol is readily adjustable to the microbial load of the processed samples without compromising protocol efficiency. Our study provides a first systematic and extensive analysis of the different options for extraction of nucleic acids from glacier-fed streams for high-throughput sequencing applications, which may be applied to other extreme environments.

Introduction

The advent of high-throughput sequencing technologies has brought hitherto inconceivable capacities to characterize the microbial ecology of both well-studied (Jansson & Hofmockel, 2018; Nielsen & Ji, 2015) and under-explored environments (Hotaling, Hood & Hamilton, 2017; Milner et al., 2017). Among the latter include high-mountain and particularly glacier-fed streams (Milner et al., 2017) and the microbial biofilms that colonize their beds (Battin et al. 2016). Today, these streams are changing at an unprecedented pace owing to climate change and the thereby shrinking glaciers, and yet little is known of their microbial diversity (Wilhelm et al., 2013; Milner et al., 2017). Glacier-fed stream (GFS) sediments are extreme habitats characterized by low microbial cell abundance and activities but very high loads of fine mineral particles (Wilhelm et al., 2013; Godone, 2017; Peter & Sommaruga, 2017; Chanudet & Filella, 2008). In order to understand the diversity and composition of these microbial communities, including both eukaryotes and prokaryotes, and the role that they play, it is essential to extract nucleic acids in sufficient quantity and quality from often complex environmental matrices. After extracting the nucleic acids, downstream applications including molecular biology methods such as PCR and next-generation sequencing of amplicons or metagenomes allow for the compositional, functional and phylogenetic characterization of microbial populations and the communities that they form (Roume et al., 2013).

While there is no lack of protocols and literature pertaining to the extraction of nucleic acids from a wide variety of environments (Roume et al., 2013; Miller et al., 1999; Xin & Chen, 2012; Porebski, Bailey & Baum, 1997; Zhou, Bruns & Tiedje, 1996), few reports dwell on the utility of these methods for biomolecular extractions from sedimentary samples with very low cell abundance as typical for GFS (Wilhelm et al., 2013; Ren et al., 2017) and in Lever et al. (2015) elaborately described diverse factors and components that need to be considered for efficient nucleic acid extractions. These include, but are not limited to key steps like cell lysis, removal of impurities and inhibitors and of critical additives like carrier DNA molecules to enhance aggregation and thus precipitation of DNA in case of very low concentrations. Since the first extraction of DNA by Swiss medical doctor Friedrich Miescher in 1869 (Dahm, 2008), biomolecule extractions have shifted from those performed with solutions prepared primarily in the laboratory (Sambrook & Russell, 2006; Miller et al., 1999) to using commercially-available kits. These ready-made options are designed to avoid the use of volatile and toxic chemicals such as phenol and chloroform, and are tailored to various environments including blood, faecal material, plant and soils (Claassen et al., 2013; Psifidi et al., 2015; Smith, Diggle & Clarke, 2003; Vishnivetskaya et al., 2014). While studies have concentrated on nucleic acid extraction from glacial ice cores (Dancer, Shears & Platt, 1997) or surface snow (Pei-Ying et al., 2012), none have demonstrated their utility for GFS sediments. Together with low cell abundance (Wilhelm et al., 2014; Lever et al., 2015), the complex mineral matrices in GFS (Peter & Sommaruga, 2017)—a consequence of the erosion activity of glaciers (Bogen, 1988)—may affect nucleic acid extraction efficiency (Lever et al., 2015). As we attempt to better understand how nature works at its limits through the study of extreme environments, non-commercial approaches (Mukhopadhyay & Roth, 1993)and methodologies (Lever et al., 2015), need to be revisited and optimized.

In recent years, several research groups (Besemer et al., 2012; Ren et al., 2017; Ren, Gao & Elser, 2017; Dancer, Shears & Platt, 1997) have successfully used kit-based methods for DNA extraction and subsequent 16S ribosomal RNA gene amplicon sequencing on GFS samples. However, the requirements for whole genome shotgun sequencing currently include at least 50 ng of input DNA to minimize bias due to PCR reactions during library preparation (Kebschull & Zador, 2015; Bower et al., 2015; Thomas, Gilbert & Meyer, 2012; Chafee, Maignien & Simmons, 2015; Peng et al., 2020). Here, we address the utility and efficiency of the “gold” standard phenol-chloroform extraction (Dairawan & Shett, 2020), and three alternative methodologies to identify the process(es) that yield not only the highest quantity but also quality of DNA, from GFS sediments. Our goal was to address whether the phenol-chloroform method yielded the expected diversity and taxonomic profiles when extracting GFS sediments, while also enabling reconstruction of metagenome-assembled genomes. Simultaneously, we wanted to validate the utility of this method for the extraction of nucleic acids from both pro- and eukaryotic sources. Overall, our findings provide a framework for the extraction of nucleic acids such as DNA for whole genome shotgun sequencing from GFS sediments, whilst highlighting the potential variability introduced due to the isolation method employed.

Methods

Sample origin & collection

DNA extraction protocols were benchmarked using three different GFS sediments from the Swiss Alps: Corbassière (CBS, collection date: 13.11.2018), 2444 m above sea level (m a.s.l) and Val Ferret (FE), 1995 m a.s.l at the glacier snout (up site, FEU, collection date: 23.10.2018) and one kilometer further downstream (down site, FED, collection date: 24.01.2019). Sampling was always performed later in the morning, before noon. Sediments (0.25 to 3.15 mm) were collected using two flame-sterilized metal sieves with a mesh size of 3.15 mm and 0.250 mm respectively. CBS differs from FEU and FED in terms of bedrock geology, with clastic sedimentary limestone dominating the catchment of CBS and brecchiated gneiss dominating in FEU and FED. Sediments generally contain more organic material further downstream from the glacier, which may inhibit DNA extraction. Sediments (0.25 to 3.15 mm) were collected using flame-sterilized sampling equipment. Wet sediments were transferred into 10 ml sterile, DNA/DNase-free tubes and immediately flash-frozen in liquid nitrogen in the field. Samples were transferred to the laboratory and kept at −80 °C until analysis. All necessary measures were taken to ensure contamination-free sampling.

DNA extraction methods

Four different DNA extraction methods were applied to the samples. The key characteristics of the different methods are summarized in Table 1. Method-1, -2 and -4 were non-commercial protocols differing primarily in the lysis step (bead-beating and lysis buffer composition; Roume et al., 2013; Lever et al., 2015; Sambrook & Russell, 2006) while method-3 was a modified protocol of the DNeasy PowerMax Soil Kit (Cat.No. 12988-10) provided by Qiagen (based on communication exchanged with the manufacturer). Due to the very low microbial abundance, additional precautions were taken to establish contamination-free conditions, including daily decontamination of equipment/areas with bleach, using DNA/DNase-free glassware and plasticware, reagents and chemicals. Additionally, “germ-free” sediment is not a viable option and is hard to remove any and all microorganisms from sediments. So, extraction blanks, i.e., tubes without any sample, were used as controls, which underwent the same extraction protocols along with the other samples. Post-DNA recovery, we assessed whether any of the eluted samples had DNA via both NanoDrop and Qubit, and found them to be below detectable levels. Additionally, the PCR reactions during library preparation did not yield any product confirming serving as a contamination-check. The input weight of sediment ranged from 0.5–5 g and are described in the respective protocols.

Table 1 Key characteristics of the four selected methods.

The table lists the key and specific characteristics of the four extraction methods tested, where n, is the total number of times each condition was tested on the material; RT: room temperature.

	Method-1
(n = 3)	Method-2
(n = 6)	Method-3
(n = 30)	Method-4
(n = 14)	
Sample prep	Lysis matrix E tube
(beads diam 1.4, 0.1, 4 mm)	0.1 mm Zirconium beads
dNTP solution	PowerBead tubes
(carnet 0.7 mm)	–	
Cell lysis	Lysis buffer
CTAB/KaPO4, pH 8
Phenol:Chloroform:Isoamyl alcohol
Bead-beating	Lysis buffer
GuHCl/EDTA/Triton X-100, pH 10
Mild vortex\Incubation at 50 °C	PowerBead Buffer+C1
Phenol:Chloroform:Isoamyl alcohol
Vortex (MO BIO vortex adapter)	Lysis Buffer
Tris-HCl, EDTA, SDS pH 8.0
Mild vortex
Incubation at 37 °C
Proteinase K addition and incubation at 70 °C	
Purification	Chloroform:Isoamyl alcohol (x1)	Chloroform:Isoamyl alcohol (x1)	Inhibitor Removal Technology (C2, C3)	Phenol:Chloroform:Isoamyl alcohol (x1)
Chloroform:Isoamyl alcohol (x1)	
Precipitation	Linear polyacrylamide
PEG-6000
Ice	Linear polyacrylamide
NaCl
Ethanol
RT	1st: Column binding & cleaning
C4, C4+EtOH, C5, EtOH
Elution: 6 ml EB
2nd: Linear polyacrylamide
NaCl, Ethanol	Linear Polyacrylamide
Sodium acetate
Isopropanol
−20 °C	

Method-1 was based on a previously established method (Griffiths, Whiteley & Bailey, 2000). Introduced modifications concerned primarily the step of mechanical lysis and DNA precipitation that was rendered more stringent to improve the recovery of small amounts of DNA. Sample cell lysis was achieved by adding 0.5 g of sediment into a lysing matrix E tube with beads of variable diameter provided by the manufacturer (MP Biomedicals, SKU 116914050), 500 µl CTAB buffer (5% CTAB, 120 mM KPO4, pH 8.0) and 500 µl of phenol/chloroform/isoamyl alcohol (ratio 25:24:1). Samples were loaded on a Precellys beater for 45 s at 5.500 r/s. DNA was extracted once more with chloroform/isoamyl alcohol (24:1) and precipitated with 2 vol PEG-6000, 15 µg/ml linear polyacrylamide (LPA) and 2 h incubation on ice (Supplemental Information).

Method-2 was an adaptation to alpine stream sediments of the modular method for DNA extraction previously published (Lever et al., 2015). The appropriate modules of the method, based on the nature of our samples, were put together in our protocol without further modification. Samples were prepared by mixing 5 g of sediment, 10–20% of 0.1 mm zirconium beads and 1 ml of 100 mM dNTP solution. Cell lysis was achieved with 5 ml lysis buffer (30 mM Tris-HCl, 30 mM EDTA, 1% Triton X-100, 800 mM guanidium hydrochloride, pH 10.0) and incubation at 50 °C for 1 h with gentle agitation (Hybridization oven, Labnet Problot L6). The supernatant was extracted once with chloroform/isoamyl alcohol (24:1) and DNA was precipitated with 10260∕280g/ml LPA, 0.2 vol 0.5 M NaCl, 2.5 vol ethanol and 2 h incubation at RT in the dark (Supplemental Information). The input weight of 5 g sediment was a modification from previously established protocol and the subsequent reagent volumes were adjusted accordingly.

Method-3 has been previously applied successfully on sand and clay soils (Hale & Crowley, 2015). In this protocol, the standard lysis capacity of the DNeasy PowerMax Soil Kit (Qiagen, Cat. No. 112988-10) was modified and enhanced by the addition of phenol:chloroform:isoamyl alcohol along with PowerBeads (kit provided) and C1 solution to 5 g of sediment. Then, the manufacturer-suggested sequence of treatments and rinses with the standard buffers of the kit were followed to reach elution of extracted DNA from silica columns with 6 ml of elution buffer. Further concentration of extracted DNA was carried out with the addition of 240 µl 5M NaCl, 2.5 vol ethanol and 10µg/ml LPA (Supplemental Information). LPA was an additional modification to the original protocol for improved DNA recovery.

Method-4 involved chemical and enzymatic treatment of samples according (Green & Sambrook, 2017) with minor modifications. Five g of sample was mixed with 10 ml of lysis buffer, incorporating the SDS as well, (0.1 M Tris-HCl pH 7.5, 0.05 M EDTA pH 8, 1.25% SDS) and 10 µl RNase A (100 mg/ml). Then sediment was vortexed for 15 s and incubated at 37 °C for 1 h in a hybridization oven. 100 µl Proteinase K (20 mg/ml) was added in a subsequent step and the mixture was incubated for 10 min at 70 °C. Samples were extracted once with phenol/chloroform/isoamyl alcohol (ratio 25:24:1) and supernatants were extracted subsequently with chloroform/isoamyl alcohol (24:1). More stringent DNA precipitation conditions were applied with the addition of 10 µg/ml LPA, and overnight incubation at −20 °C (Supplemental Information).

All DNA extracts were suspended in 100 µl of DNA/DNase–free water (ThermoFisher Scientific). Due to the inadequacy of DNA obtained from Method-1 given the 0.5g input sediment weight, we scaled the extraction to 5 g starting weight prior to sequencing. Extracted DNA was thereafter stored at −20 °C until further use. Due to the low DNA yields, it was necessary to use Qubit dsDNA HS kit (Invitrogen), a fluorescent DNA quantification method with high sensitivity. Quality assessment, with Nanodrop and DNA visualization on 0.8% agarose gel containing GelRed nucleic acid stain, was possible only for DNA extracted with method-4 and for DNA concentrations higher than 0.5 ng/ul. All samples yielded sufficient DNA, i.e., 50 ng (total input), for metagenomic sequencing and subsequent analyses. Additionally, a commercially-available microbial mock community (ZymoBIOMICS, Cat.No. D6300) was extracted using Method-4 and used for subsequent sequencing.

DNA sequencing

50 ng of DNA from all samples were subjected to random shotgun sequencing. The sequencing libraries were prepared using the NEBNext Ultra II FS DNA Library Prep kit for Illumina (Cat.No. E7805) using the protocol provided with the kit. The libraries were prepared considering 350 base pairs (bp) average insert size. Qubit (Invitrogen) was used to quantify the prepared libraries while their quality was assessed on a Bioanalyzer (Agilent). We used the NextSeq500 (Illumina) instrument to perform the sequencing using 2 ×150 bp read length at the Luxembourg Centre for Systems Biomedicine Sequencing Platform.

Genome reconstruction and metagenomic data processing

Paired sequences (i.e., forward and reverse) were processed using the Integrated Meta-omic Pipeline (IMP) (Narayanasamy et al., 2016). The metagenomic workflow encompasses pre-processing (read quality filtering and trimming), assembly, and genome reconstruction in a reproducible manner. The adapter sequences were trimmed in the pre-processing step including the removal of human reads. Thereafter, de novo assembly was performed using the MEGAHIT (version 2.0) assembler (Li et al., 2015). Default IMP parameters were retained for all samples. Subsequently, we used MetaBAT2 (Kang et al., 2019) and MaxBin2 (Wu, Simmons & Singer, 2016) for binning in addition to an in-house binning methodology previously described (Heintz-Buschart et al., 2017). The latter method initially ignores the ribosomal RNA sequences in kmer profiles based on VizBin embedding clusters (Laczny et al., 2015). In this context, VizBin utilises density-based non-hierarchical clustering algorithms and depth of coverage for genome reconstructions. Subsequently we obtained a non-redundant set of metagenome-assembled genomes (MAGs) using DASTool (Sieber et al., 2018) with a score threshold of 0.7 for downstream analyses. The abundance of MAGs in each sample was determined by mapping the reads to the reconstructed genomes using BWA-MEM (Li, 2013), taking the average coverage across all contigs. Diversity measures from metagenomic sequencing were assessed by determining the abundance-weighted average coverage of all the reads to identify the number of non-redundant read sets (Rodriguez-R & Konstantinidis, 2014).

Taxonomic classification for metagenomic operational taxonomic units

We used the trimmed and pre-processed reads from the IMP workflow to determine the microbial abundance and taxonomic profiles based on the mOTU (v2) tool (Milanese et al., 2019). Based on the updated marker genes in the mOTU2 database including those from the TARA Oceans Study (Sunagawa et al., 2015) and recently generated MAGs (Tully, Graham & Heidelberg, 2018), taxonomic profiling was performed on our sequence datasets. We used a minimum alignment length of 140 bp to determine the relative abundances of the mOTUs, including the normalisation of read counts to the gene length, also accounting for the base coverage of the genes. Additionally, we used CheckM (Parks et al., 2015) to assess completeness and contamination. Subsequently, taxonomy for MAGs recovered after the redundancy analyses from DASTool was determined using the GTDB (Genome Taxonomy Database) toolkit (gtdb-tk) (Parks et al., 2018).

Data analysis

All figures for the DNA concentrations, library preparation, assembly metrics and supplementary figures were generated using GraphPad Prism (v8.3.0). Taxonomical assessment and diversity measures were created using version 3.6 of the R statistical software package (R Core Team, 2013). DESeq2 (Love, Huber & Anders, 2014) with FDR-adjustments for multiple testing were used to assess significant differences in the MAG abundances. The genomic cluster figure for the mock community was obtained as an output from the IMP metagenomic pipeline.

Results

Phenol-chloroform-based extraction method results in higher DNA yields

To ensure native sequencing, by minimizing the number of PCR (polymerase chain reaction) cycles within the library preparation protocols, we tested four protocols for biomolecular extraction, with an aim of acquiring large quantities (>50 ng) DNA from glacier-fed stream benthic sediments. The four methods tested were selected because of their wide applicability on related environmental samples (Method-1 & -2) (Griffiths, Whiteley & Bailey, 2000; Lever et al., 2015; Tatti et al., 2016) and their improved chances of higher yields (Method-3; Qiagen communication). Since method-4 is considered the gold standard of DNA extraction in biomedical sciences (Dairawan & Shett, 2020) and bacterial cultures (Green & Sambrook, 2017), it was included in our study. The four protocols are largely based on the same principles, viz. sample preparation, cell lysis, purification, precipitation and washing (Table 1). From preliminary tests, it became apparent that a small-scale approach (Method-1; 0.5 g input sediment) did not yield sufficient amounts of DNA for metagenomics due to, on average, limited microbial biomass in the samples. Thus, all protocols (aside from Qiagen’s - already produced for maxi scale) were scaled up to 5 g of input sediment and a co-precipitant, like linear polyacrylamide, was included in all precipitation steps. This was essential for the quantitative recovery of the small amounts of extracted DNA from high solution volumes (6–10 ml).

Overall, we found that extractions using the commercial kit from Qiagen (method-3) yielded increased total DNA as compared to a commonly used protocol (method-1; Fig. 1A). Furthermore, method-3 was similar in terms of DNA yield when compared to a generalized protocol (method-2) previously proposed (Lever et al., 2015) (Fig. 1B). On the other hand, the phenol-chloroform based extraction protocol (method-4) was tested against both methods 2 and 3, using sediment samples collected from the three different glacier floodplain streams (CBS, FEU, FED) from Switzerland. Method-1 was omitted from these tests due to insufficient DNA concentrations in the preliminary extractions. We found that for all three GFS, the phenol-chloroform extraction yielded the highest DNA concentrations. In some cases, and notably samples with low cell abundance, we even obtained one order of magnitude more DNA (Fig. 1C).

Figure 1 Total DNA concentrations using different extraction protocols..

Boxplots represent the total amount of DNA (ng) extracted from 5 g of sediment when comparing (A) method-1 versus the modified-commercial kit-based method-3 and (B) method-2 versus method-3. (C) Boxplots of the DNA quantities isolated from three glacial floodplains (CBS - Corbassière, FEU - Val Ferret up site, FED - Val Ferret down site), using method -2, -3 and -4. Method-1: CTAB buffer lysis (Griffiths, Whiteley & Bailey, 2000), Method-2: Modular DNA extraction (Lever et al. 2015), Method-3: Qiagen PowerMax Soil DNA extraction kit, Method-4: Chemical and enzymatic extraction. Significance was tested using a Two-Way ANOVA with Student-Neuman Keul’s post-hoc analyses. ∗∗p < 0.01, ∗∗∗p < 0.001.

Quality assessment of these DNA extracts with Nanodrop showed OD260/280 ratios between ∼1.4 and ∼1.6. Agarose gel electrophoresis revealed a high-molecular weight band with no apparent shearing, smearing or residual RNA, indicative of high-quality DNA (Fig. 2A). A secondary effect appearing in certain samples, but without any perceived consequences in the quality of extracted DNA whatsoever, was the development of a pink-red color of varying intensities with the addition of phenol:chloroform:isoamyl alcohol (Fig. 2B). This was pH dependent since samples were decolorized with the addition of sodium acetate pH 5.2 in the precipitation step. This could possibly be due to a ferric-chloride-phenol compound formed when chloride and phenol constituents of the protocol interact reversibly with Fe+3 ions contained in certain samples depending on local geology (Banerjee & Haldar, 1950). Similar coloration has been previously reported (Lever et al., 2015).

Figure 2 Characteristics of DNA extracted with method-4.

(A) Agarose gel electrophoresis of DNA extracted with mild vortexing of sediments and incubation in lysis buffer, proteinase K treatment and phenol-chloroform extraction. Lane 1: GeneRuler 1 kb DNA ladder; lanes 2-4: CBS, FED, FEU respectively. (B) Pink-red supernatants developed during phenol:chloroform extraction step.

Extraction method affects library preparation efficiency

The DNA extractions based on method-3 and using phenol-chloroform methods were subsequently subjected to library preparation for high-throughput whole genome shotgun sequencing. Despite similar quality of DNA across both methods (∼1.4–1.6 OD260/280), library preparation using the modified commercial kit did not yield any successful libraries (Fig. 3). To assess if any impurities or inhibitors hampered library preparation we tested two clean-up methods for the DNA extracted from the commercial kit: (1) ethanol precipitation and (2) magnetic-bead based clean-up. For magnetic bead clean-up the SPRIselect beads (Beckman Coulter, 23318) were used according to the manufacturer’s protocol. We found that the magnetic-bead method leads to a complete loss of sample (i.e., undetectable DNA quantity via Qubit analyses) during the process, especially if starting with a low input DNA concentration. Although we lost six out of twelve samples using the magnetic-bead clean-up, we achieved 100% efficiency as indicated by a concentration of greater than 0.5 ng/ul after library preparation quantified by Qubit, with the remaining six samples. On the other hand, ∼20% of the samples cleaned via ethanol precipitation failed library preparation. Contradictory to these methods, DNA extracted using the phenol-chloroform based method (method-4) yielded 100% efficiency in terms of library preparation without any additional clean-up (Fig. 3). Additionally, we found that the distribution of the total yield after library preparation using the phenol-chloroform method was more uniform across samples compared to the other methods (Fig. 3).

Figure 3 Library preparation efficiency.

The efficiency or success percentage for prepared libraries based on the individual methods is indicated in the table. Boxplots represent concentrations of the prepared libraries.

Whole genome shotgun assembly unaffected by extraction methods

Extraction methods for whole genome shotgun sequencing may affect the sequencing itself, including the quality and assembly of the reads downstream. To assess this, we used the libraries prepared as described above (Fig. 3), and performed whole genome shotgun sequencing on an Illumina NextSeq500. The average quality across all three methods based on short-read sequencing was Q30 after trimming the leading and trailing sequences (described in Methods). We assessed several assembly metrics including the average length of contigs (N50), largest alignment, total aligned length and coverage. We did not find any significant differences among any of these measures across all three methods (Figs. 4A–4C, 4E). Using a diversity index metric, we however found a more uniform distribution across all samples prepared using method-4, albeit no significant differences to the commercial kit-based extraction and library preparation (Fig. 4D).

Figure 4 Estimate of assembly metrics following extraction.

Barplots demonstrate the (A) N50 for the sequence assemblies, (B) length of the longest aligned sequence, (C) the total aligned length. Bars indicate standard deviation from the mean. (D) Boxplot showing the nonpareil diversity index across the three groups. (E) Percentage of coverage of the assembled sequences by read-mapping is depicted.

Extraction methods influence metagenomic profiles

It is well established that extraction methods (Mackenzie, Brett & Taylor, 2015) and library preparation (Bower et al., 2015) protocols affect the taxonomic profiles and genomes recovered after high-throughput sequencing. We determined if the preparation methods affected the overall diversity of taxa recovered and found that phenol-chloroform and the magnetic-bead clean-up methods demonstrated similar levels of diversity (Shannon) as compared to samples precipitated using ethanol (Fig. 5A). Overall, the community profiles of the ethanol precipitation-based method were highly diverse (Fig. 5B). Interestingly, the genomes recovered and their abundances were similar in the phenol-chloroform and magnetic-bead methods as well (Fig. 5C). However, we observed a significant increase (p < 0.001, FDR-adjusted p-value) in the abundance of a Ralstonia genome when prepared with the ethanol precipitation protocol (Fig. S1). Additionally, we found that the number of genomes recovered using the phenol-chloroform was more consistent with previously reported 16S rRNA gene sequencing profiles for GFS from Austria (Wilhelm et al., 2013; Besemer et al., 2012; Wilhelm et al., 2014). Simultaneously, we used an approach to identify metagenomic operational taxonomic units (mOTUs) and found that the phenol-chloroform and magnetic-bead methods showed similar profiles of mOTUs compared to that of ethanol precipitation (Fig. 5D).

Figure 5 Diversity and taxonomic profiles of the metagenomic sequencing.

(A) Boxplot showing the Shannon diversity index for the taxonomic profiling for the three groups. Significance was tested using a One-way ANOVA with Student-Neuman Keul’s post-hoc analysis. ∗∗∗p-value <0.001, ∗∗∗∗p-value <0.0001. (B) Principal component analyses generated using Bray–Curtis dissimilarity matrix depicts similarities or lack thereof between the three groups. (C) Abundances of the reconstructed genomes are depicted for method-3 + EtOH, method-3 + magnetic bead clean-up and method-4 extraction. (D) Heatmap demonstrating the mOTUs for the three methods is depicted. The hierarchical clustering for the heatmap was generated using Ward’s clustering algorithm.

Efficiency of phenol-chloroform extraction on a mock community including eukaryotes

To determine whether the phenol-chloroform extraction method is biased against eukaryotes, we used a commercially-available mock community (ZymoBIOMICS Microbial Community Standard #D6300) to assess bias and errors. After sequencing, we recovered high quality (>90% completion, <5% contamination) bacterial genomes (Fig. 6A). Additionally, the abundance of the microbial genomes, including one of the eukaryotes - Saccharomyces cerevisiae, were similar to the expected levels in the mock community (Fig. 6B). On the other hand, the protocol enabled the identification and partial recovery of the Cryptococcus neoformans genome, albeit at lower levels possibly due to increased melanisation of the cell wall (Grossman & Casadevall, 2017) affecting lysis and subsequent extraction.

Figure 6 Evaluation of phenol-chloroform extraction using a mock community.

(A) Scatterplot depicts the clusters of contigs representative of the reconstructed genomes after processing the mock community using the IMP meta-omics pipeline. The taxonomic identity is displayed next to the respective clusters. (B) Barplots indicate the relative abundance of the individual genomes recovered from the mock community sequencing after extraction with the phenol-chloroform method. The upper (black) line represents the expected abundance (12%) of the prokaryotes, while the lower (red) line indicates the expected abundance (2%) of the eukaryotes.

Discussion

Improved omic techniques not limited to metagenomics are robust methods for analyzing nucleic acids and the characterisation of microbial communities in various environments (Jansson et al., 2012). One way of understanding the impacts of global climate change on GFS includes the establishment of their census of microbial life (Milner et al., 2017). However, methods designed for the extraction of biomolecules including DNA have not been validated for GFS sediments. Although previous glacier-fed streams studies successfully used extracted DNA for 16S rRNA amplicon sequencing (Ren et al. 2017; Ren, Gao, and Elser 2017; Vardhan (Reddy & Vishnu, 2009; Wilhelm et al., 2013) the input DNA concentration requirements are considerably higher for whole genome shotgun sequencing. In order to pursue a deeper characterisation of the microbial communities within the GFS sediments, increased concentrations of DNA may additionally alleviate PCR biases (Brooks et al., 2015; Kim & Bae, 2011). Also, as previously highlighted, several methods exist for extractions from a wide variety of environmental samples, but not for GFS sediments. Here, we systematically tested the utility of four extraction protocols to identify a ubiquitous methodology. We found that a phenol-chloroform based extraction protocol can be used for samples across geographical separations, differences in bedrock, and samples collected at various distances from the glacier.

Glassing et al. (2016) demonstrated that inherent DNA contamination may influence microbiota interpretation in low biomass samples. Additionally, it is known that certain compounds—polysaccharides, humic acids, may affect PCR reactions Rådström et al., 2004, requiring the need for additional DNA clean-up. It has been established that DNA losses occur during the purification steps (Roose-Amsaleg, Garnier-Sillam & Harry, 2001), including when using commercial column methods (Howeler, Ghiorse & Walker, 2003; Lloyd et al., 2013), and phenol-chloroform (Ogram, Sayler & Barkay, 1987). Interestingly, we found similar losses when using the magnetic bead clean-up, whereas the ethanol precipitation method was inefficient compared to the phenol-chloroform protocol. Though the kit-based methods are more convenient and safer than phenol-chloroform extractions (Tesena et al., 2017), access to reagents and costs may be a considerable factor. On the other hand, isolation of the aqueous phase from phenol-chloroform can be user-dependent potentially affecting reproducibility, while kits have been shown to be more consistent across samples (Claassen et al., 2013). Another key feature of our findings was the potential for the kit-based methods to influence the efficiency of genome reconstruction and variability in the taxonomic profiles that were recovered. While this has been reported previously (Mackenzie, Brett & Taylor, 2015; Carrigg et al., 2007), we found considerable variability when compared to phenol-chloroform. This is plausible due to the incomplete dissolution of DNA in buffers, especially when using methods involving charged minerals (Vorhies & Gaines, 2009; Barton, Taylor & Pemberton, 2006; Vishnivetskaya et al. 2014), which may additionally affect DNA stability.

Conclusions

The utility of extraction methods extends beyond the process itself, impacting downstream applications such as whole genome shotgun sequencing. Our study shows that phenol-chloroform may be an under-appreciated yet powerful method for isolating nucleic acids from glacier-fed stream sediments. While additional steps may be required towards extraction of other biomolecules such as RNA, proteins and metabolites, minor modifications may be sufficient (Toni et al., 2018). Moreover, we report for the first time a systematic assessment of biomolecular extraction methods for GFS sediments. Our findings though fundamental and previously unexplored, may lay the foundations for future in-depth characterisation of GFS microbial communities.

Supplemental Information

Supplemental Information 1 Supplementary Information and Protocols

Four extraction methods used and validated in the study are described in detail.

Click here for additional data file.

Supplemental Information 2 Relative abundance of Ralstonia sp. AU12-08

The abundance of the Ralstonia genome recovered from the samples when processed with method-3 (EtOH and magnetic bead clean-up) and method-4. Significance was tested using One-Way ANOVA with Student-Neuman Keul’s post-hoc analyses. ∗∗∗p < 0.001,  ∗∗∗∗p < 0.0001.

Click here for additional data file.

We are grateful to Laura de Nies, Camille-Martin Gallausiaux, Jean-Pierre Trezzi, Cedric Laczny, Audrey Frachet, Lea Grandmougin, Annegrat Daujemont, Laura Lebrun (LCSB) for discussions and laboratory support. The University of Luxembourg Sequencing Platform and HPC facilities were highly instrumental for the in-silico analyses.

Additional Information and Declarations

Competing Interests

Author Contributions

Data Availability

The authors declare there are no competing interests.

Susheel Bhanu Busi and Paraskevi Pramateftaki conceived and designed the experiments, performed the experiments, analyzed the data, prepared figures and/or tables, authored or reviewed drafts of the paper, and approved the final draft.

Jade Brandani, Stilianos Fodelianakis, Hannes Peter and Rashi Halder performed the experiments, authored or reviewed drafts of the paper, and approved the final draft.

Paul Wilmes and Tom J. Battin conceived and designed the experiments, authored or reviewed drafts of the paper, and approved the final draft.

The following information was supplied regarding data availability:

Data is available at NCBI: PRJNA624048.

https://www.ncbi.nlm.nih.gov/bioproject/PRJNA624048/.

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
