# Peer review of "Optimised biomolecular extraction for metagenomic analysis of microbial biofilms from high-mountain streams"

_PeerJ, doi:10.7717/peerj.9973_

## Round 0.1 · original submission · Major Revisions

Thank you for submitting your manuscript to PeerJ. While the global pandemic made finding reviewers a bit slower than normal, we were able to find three experienced reviewers familiar with the topic and the rationale for the study, and all three provided both timely and thorough reviews, which I greatly appreciated. I am pleased that all three reviewers were complimentary of the science and the writing on this manuscript. All three of the reviewers provided extensive suggestions for improvement, and we request that along with your revised manuscript you provide detailed notes on how you addressed each comment with edits and line numbers in the manuscript...ideally copying the revisions into the response document to facilitate rapid review by the reviewers and me.

Reviewer 1 ·

Basic reporting

An interesting and important study! This manuscript was well written and organized. I do like the work, but I think this manuscript would benefit from revisions on figures, especially.

Experimental design

The experiments were well designed.

Validity of the findings

Generally good. The figures can be improved a little bit to clarify the results. Referring to the comments below.

Additional comments

L22: “GFS ecosystems”
L114: add “,” after "abundance"
L135-150: some of the tenses were wrong, which should be past tense.
L218: delete “of”
L239: delete “in”
L252: change “extracted” to “extractions”
L278: delete “were observed compared”
Figure 1: it is better to add the significant difference results in the plot using different letters, stars, or just P-values, or like Figure 6a. Moreover, it would be good to briefly explain what the methods -1,2,4 are, so the readers can understand the figure without referring to the main context.
Figure 2: briefly explain method-4.
Figure 3: is this figure really necessary? Or combine this figure with others?
Figure 4: it is weird to have a table in a figure. It is easy to convert the table as annotations in the figure.
Figure 5: (1) there seems no reason to use different plots in figure with boxes and bars together. You can use either one. (2) the variables’ name is better to show on the top of the plot not on the X-axis like Figure S1. The legends (extraction methods) can be added to the X-axis.

Reviewer 2 ·

Basic reporting

Manuscript # 48507
Title: Optimised biomolecular extraction for metagenomics analysis of microbial biofilms from high-mountain streams.
Authors: Susheel Bhanu Busi, Paraskevi Pramateftaki, Jade Brandani, Stylianos Fodelianakis, Hannes Peter, Rashi Halder, Paul Wilmes, Tom Battin

 The manuscript is well written - clear, professional and articulate English has been used throughout. In microbial ecology, we still face several challenges in the area of finding/devising efficient DNA extraction methods to study microbial community dynamics in understudied locations, and therefore studies such as this merit the scope of enhancing our opportunities to explore remote and less-represented sites such as the ones represented in this study.

I have the following comments about the manuscript, please see below.

INTRODUCTION:
 Overall, I see a lack of citations throughout the introduction of the manuscript. While authors make several statements to support the goal of their study, they must also cite enough sources to back up their claims. Please see a few suggestions below:

• Line 42: Please add a few citations for “under-studied environments”
• Line 46-48: Are there studies/reviews that more descriptively talk about this? Please include a few citations not just to support this statement, but also to help readers refer back to more detail on this should they wish to.
• Line 73-75: Please add citation(s).
• Line 75-77: add citations.
• Lines 82-83: needs a reference, I am aware of this 50 ng cutoff, there must be some literature on this, just cite a few here.

 Lines 86-93: There’s no need to include results in introduction. Authors should rephrase this paragraph so that this reads to include the goals of your study based on the background and rationale they’ve have provided earlier in the introduction. Results should go under the separate section.


METHODS:
 Lines 98, 99, 100 – What is CBS, FE, FEU, FED? Please elaborate of abbreviations on first use.
 Line 98: What is a.s.l.? Please elaborate on first use.
 What are the dates and times of sampling? Please add this to sample collection methods section.
 Line 103: What kind of sampling equipment was used?
 This is a comment for ALL the four DNA extraction protocols the authors have included under the methods section: it seems like all of these protocols have been adapted from already existing protocols or commercially-available kits. Authors mention that they’ve modified these existing methods. Since this is methods paper, it is very important to specify in detail the steps that have been specifically modified in your own procedure.

 Line 130: Mention the equipment used for agitation.

 This is also a comment for ALL the extraction methods: Did the authors start with the same weight of sample for all extractions? If not, a systematic and scientific comparison of the methods is not possible. Authors mention between lines 222-224 and in Figure 1 that 5 g was used for all. In the methods, maybe a range should be used. In line 119, it is mentioned that 0.5 g of sediment was the starting sample, whereas in line 128, it is mentioned that 5 g of sediment was the starting material. While there’s no mention of the amount for method 3, method 4 in line 144 again states that 5 g of sediment was the starting material.

 What kind of normalization was achieved for this comparison? In other words, what amount of DNA was the starting material for the DNA sequencing from each of the methods? Authors should mention this in their methods. You could add a line or two in line 158 to address this issue. As this is a methods paper, simply saying ‘sufficient” DNA was obtained is not enough. What were the concentrations obtained based on the starting material? This is could also be provided as a short table to enhance the understanding for the readers.

 How did the authors address contamination issues? For a methods paper concerning DNA extraction, this must be included in methods for readers who might be following your protocols later.

 What were the controls used during the extraction procedures? Please add this to methods.

 Figures 2 and 3 are images, they could be combined together to form one single figure. This helps to make the manuscript concise and precise.

 Figure 2 shows the bands from only one of the extraction methods. Could the authors provide a comparative yield image of all four methods that has been started with the same amount of sediment?

Experimental design

please see my entire review under BASIC REPORTING section

Validity of the findings

please see my entire review under BASIC REPORTING section

Reviewer 3 ·

Basic reporting

line 42- sentence starting "among the latter" is awkward. Perhaps add "Among the latter include..."



otherwise all is OK.

Experimental design

line 119 - please clarify - these were the beads that are already in that tube? Please provide manufacturer or kit name, only SKU number is listed.

line 136 - list manufacturer of power soil kit. (note, this kit has changed components when MoBio sold it so be clear as to which company it came from)

line 138: what is meant by "elaborate"? Are you just following kit protocol?

line 143 - you switch tenses here. All other methods were past tense.

line 154 - use a method "like the Qubit" or did you actually use a Qubit

line 173: when was quality control of the reads performed?

at the end of methods, you mention a "mock community" yet the methods to create this mock community are not described.

In the results, line 257 you describe a magnetic bead clean up, but this was not clear in the methods.

line 300 - how exactly did you utilize this mock community? Did you add it to sediment and then extract? This needs a section in the methods.

line 347 - please show the data. Considering it comes this late in the manuscript, you could even leave this sentence out.

Figure 3 can be moved to supplemental. It is not informative to your findings.

Figure 4- I'm confused on what 100% success means. Please clarify your units. Do you mean micrograms in resulted in the micrograms out of library prep?

Please provide more detailed information on exactly how many times each condition was tested on the same material.

Validity of the findings

All findings seem valid, details just need to be more clearly explained.

---

## Round 0.2 · accepted · Accept

Both reviewers have agreed and I agree that your revisions were thorough and complete. We are pleased to accept your manuscript.

Reviewer 2 ·

Basic reporting

The authors have made all changes based on my previous review suggestions.

Experimental design

NA

Validity of the findings

NA

Reviewer 3 ·

Basic reporting

no comment

Experimental design

no comment

Validity of the findings

no comment

Additional comments

looks good!